# Self-Perception of Inclusion in an Inclusive Choir: An Analysis Using the Scale for the Assessment of Inclusion (SAI)

**DOI:** 10.3390/bs13090758

**Published:** 2023-09-12

**Authors:** Borja Juan-Morera, Icíar Nadal-García, Belén López-Casanova, Eva Vicente

**Affiliations:** 1Department of Musical, Plastic and Corporal Expression, Faculty of Education, University of Zaragoza, 50009 Zaragoza, Spain; iciarnad@unizar.es (I.N.-G.); belocasa@unizar.es (B.L.-C.); 2Department of Psychology and Sociology, Faculty of Education, University of Zaragoza, 50009 Zaragoza, Spain; evavs@unizar.es

**Keywords:** choral singing, social inclusion, socio-musical, inclusive education, disability, diversity, Cantatutti, social transformation, inclusivity, Index for Inclusion

## Abstract

Inclusion promotes equal opportunities, and aims to eliminate discrimination, by ensuring full access, participation, and representation for all individuals in society, with music playing a crucial role in addressing this global challenge, and fostering positive and inclusive change. The aim of this study is to identify perceptions of inclusive culture, policies, values, and practices in one specific inclusive choir in Spain. The sample consisted of 135 members, ranging from 18 to 79 years of age, of which 22.2% have recognised disabilities. All of them filled in the adapted Scale for the Assessment of Inclusion (SAI) form. The results show that, regardless of gender, age, and length of participation in the choir, their members share positive perceptions of the inclusiveness of its values, policies, practices, and culture. However, perceptions vary across the groups surveyed, and are generally more favourable among participants with a disability, those without a university education, or those aged 26 or over. It is found that people who participate in an inclusive choir, in which music is the mediating resource, perceive a high level of inclusion that allows them to feel they belong to a group where diversity, equality, and the promotion of people are respected. The findings are analysed, considering the importance of conducting multidimensional evaluations that include all members of an organisation.

## 1. Introduction

The term inclusion refers to the full integration of all members of a society or community, regardless of their personal characteristics or circumstances. The term initially emerged in the field of education to describe the integration of students with disabilities into mainstream schools [1]. It has evolved from the concept of “integration” [2], and has been extended to different areas, such as labour, cultural, and social inclusion.

Inclusion is based on the principle that all individuals have the right to receive a quality education, and to participate actively in society, regardless of their disability or other characteristics. It aims to promote the participation and recognition of all groups on equal terms, eliminating all forms of discrimination.

In the context of the European Union, the term “social inclusion” emerged in the first decade of the 21st century, addressing the need to eradicate poverty and social exclusion [3]. Although the exact definition of social inclusion as the antithesis of social exclusion has been debated [4], the concept has been broadened, to encompass diverse dimensions such as race, gender, disability, sexual orientation, age, and religion.

Inclusion can manifest itself at different levels, including access to services and re-sources, participation in activities, and representation in governance and decision-making bodies [5]. It aims to ensure equal opportunities and the full exercising of rights for all people, contributing to the building of fairer and more equitable societies [6].

According to [7], equity plays a fundamental role in reducing inequalities in key outcomes, such as life expectancy, access to education, health, and political freedoms. It is seen as a crucial factor in achieving equality of opportunity for all individuals, especially those belonging to disadvantaged and poor groups.

In accordance with [8], over the last thirty years, international initiatives have been carried out to promote inclusive education, which has become a global phenomenon of great interest to the academic community, and to society in general. Inclusive education is understood as a global challenge that involves various actors, such as governments, schools, teachers, students, families, and society in general. It is a complex and multifaceted phenomenon that needs to be approached and understood in a holistic and comprehensive way. It is not limited only to the inclusion of students with disabilities, but also encompasses the inclusion of students from different ethnic, cultural, linguistic, religious, and socio-economic backgrounds.

International initiatives in inclusive education date back to the 1990 UN World Conference on Education for All [9], and the 1994 World Conference on Special Needs Education [1], which promoted the idea of an educational environment for all children. The term “inclusive education” first appeared in 2000 in the Dakar Framework for Action document [10]. Since then, instruments such as the Index for Inclusion have been developed, to support work towards inclusive education in schools.

Inclusive education has evolved over time, broadening its focus beyond students with disabilities, to include all students. Educational inclusion is recognised as a principle based on equality, justice, and equity, and its effective implementation can be influenced by political, social, economic, and cultural factors [11].

The UN’s 2030 Agenda for Sustainable Development, approved in the Spanish Congress of Deputies on 12 December 2017 [12], has set targets to ensure inclusive and equitable quality education. It highlights the importance of listening to the voices and testimonies of learners in the development of inclusive education, as well as the need to collect data to assess progress towards inclusion [13]. The implementation of inclusive education is challenging but is seen as fundamental to achieving inclusive and democratic societies.

In 2000, Ref. [14] introduced the first version of the Index for Inclusion, an assessment and development tool designed to support educational institutions, including schools and other centres, in their work towards the achievement of inclusive education. The resource was developed by the Centre for Studies on Inclusive Education (CSIE) in the UK, and has been used in over 30 countries. The Index for Inclusion is based on the premise that inclusion is not just a goal in itself, but a means to improve the quality of education for all learners. Its approach provides educational institutions with a framework that enables them to reflect on their practice, and to identify and overcome barriers to participation and progress for all students. Encompassing more than 100 indicators divided across four dimensions—context, policy and practice, participation and learning, and outcomes—the Index for Inclusion serves as a holistic and exhaustive tool.

The Index for Inclusion has proven to be a valuable resource in education, facilitating the creation of inclusive environments in schools. However, according to [15], its application to organisations providing services to people with disabilities has been limited. The authors have, therefore, developed the Scale for the Assessment of Inclusion (SAI), based on this tool.

These authors state that their adaptation of the Index for Inclusion has been constructed taking into account the specific needs of people with disabilities, and the particularities of the organisations that serve them. To this end, modifications and adjustments have been made, to adequately address the dimensions of inclusion that are relevant to this context. The authors have developed a scale with strong psychometric properties. This scale evidences the appropriateness of the adaptation made, as well as the relevance of the changes introduced. The validity and reliability of the scale were rigorously assessed, ensuring its usefulness and applicability in the context of organisations providing services to people with disabilities.

This comprehensive instrument (SAI) has been used in this study as a methodological backbone, as it provides a solid framework for the study of the self-perception of inclusiveness in the context of inclusive educational practice.

Consequently, this article disseminates the outcomes derived from the implementation of a questionnaire adapted from the Scale for the Assessment of Inclusion (SAI) in a non-formal inclusive music practice through choral singing, conducted at the Faculty of Education of the University of Zaragoza, Spain [16]. The main objective of the study was to find out and detect the perception of inclusion of people, with and without functional diversity, who participate in the Cantatutti Inclusive Choir. This research endeavor seeks to address a notable gap in the literature, as there is a dearth of comprehensive studies delving into this specific topic within the socio-musical realm. While certain projects are labeled as inclusive, a significant void exists concerning the quantification and measurement of such inclusiveness. Additionally, the linkage of these projects with a recognised framework, such as the Index for Inclusion, remains largely unexplored in the existing literature.

The Inclusive Choir Cantatutti is a musical ensemble established in Zaragoza, active since 2017, whose purpose is to foster the inclusion and active participation of individuals who are diverse in terms of ability, (dis)ability, origin, socio-economic status, culture, age, or education level. Its main objective is to provide an inclusive musical experience, where diverse people have the opportunity to sing together, and share their appreciation for music. Through this initiative, the choir seeks to challenge existing barriers and stereotypes, by creating an environment in which all participants feel recognised and respected in equal measure.

This choir is characterised by its multimodal expression, in which voice, body language, and Spanish Sign Language (LSE) are integrated, allowing the participation of people with and without functional diversity [17]. Choral practice, in this context, favours personal and collective development, by promoting self-esteem, the construction of identity within the group, friendship, and the generation of new social relationships. Participation in this choir entails the acquisition of interpersonal interaction skills, which implies a significant sense of social responsibility [18].

The participation in a project such as an inclusive choir addresses four of the SDGs of the 2030 Agenda, as it ensures and fosters the wellbeing of its members (SDG 3), promotes inclusive, equitable, and quality education and learning opportunities for all (SDG 4), contributes to a reduction in social inequalities (SDG 5), and fosters collaboration to build a peaceful and inclusive society (SDG 16).

Music is an art that has been used since ancient times as a means of expression and communication, and can be a tool to promote social inclusion and community cohesion. According to [19], the performing arts have the potential to foster active participation, social inclusion, and empowerment.

Neurological, cognitive, and social psychological research suggests that participation in musical activities can have a significant impact on social inclusion, understood as a subjective sense of belonging, and integration within a social group [20].

According to [21], music, as an artistic expression, has the potential to promote social inclusion in a community, through overcoming barriers that separate people, and thus contributing to the construction of inclusive, supportive, and discrimination-free societies. Socio-musical projects can be a valuable tool for fostering social inclusion, as they allow people of different backgrounds, ages, and musical abilities to participate in collective musical activities, creating inclusive and safe spaces for learning and participation. In addition, music can be used as a means to raise awareness, and sensitise society to inclusion and diversity issues.

Inclusive educational practice must recognise human musicality as essential to welbeing, and promote an enriched and broader curriculum that allows everyone the opportunity to be musical [22].

Inclusion in the artistic context can be approached in two different ways: guaran-teeing equal access to artistic training and opportunities, removing barriers, and allowing the full participation of all people, regardless of their personal characteristics or profiles [23,24]; and focusing on the representation of participants, which seeks to ensure that diverse perspectives and voices are reflected in artistic production, and in the cultural sphere in general.

Defining and measuring inclusion is a challenging but crucial task, as its meaning varies in different contexts. According to [25], inclusive musical experiences encompass intellectual, social, and affective processes that can bring about positive change in our society. These experiences depend on positive interactions with other individuals, as argued by [26].

Ref. [27] states that art plays a fundamental role in social reconstruction, as it facilitates individual and community integration. Moreover, artistic practice has a direct influence on increasing people’s confidence and motivation. Ref. [28] stresses that socio-educational and community projects based on music as a mediation tool can contribute to mitigating social dysfunctions, promoting cohesion, as well as a sense of belonging and community identity.

Recent studies have investigated the benefits of participating in socio-musical projects as a means of social inclusion and transformation [18,29,30]. These practices, which have been developed since the late 20th century, examine the positive impact that musical experiences and practices have on inclusion. Furthermore, the psychological and social benefits of choral singing in socio-musical projects have been evidenced for both participants and audiences alike [31,32,33,34]. These studies support the psychological and social benefits of music in general, and choral singing in particular, as well as their ability to promote social inclusion.

According to [35], the analysis of the degree of inclusion in centres and services that care for people with diverse educational needs is a relevant aspect to consider. Although there are numerous studies focusing on inclusive practices in compulsory education, the research on these practices in non-formal or adult education is limited. Consequently, this study aims to assess the level of inclusion in a choral grouping characterised by its inclusive approach, which seeks the participation of individuals from diverse backgrounds. The aim is to analyse the possible impact of different variables on the perception of inclusion in such a context. The study also tries to find out whether, as a group, its members perceive that they are part of an inclusive community, with inclusive values, policies, and practices. The aim is also to investigate whether personal variables, such as sex, gender, age, having or not having a disability, special educational needs, and/or health problems, influence perceptions of inclusion. It also aims to analyse the influence of variables more closely linked to the choir, such as seniority, sense of permanence in the institution, or the role played in it.

The hypotheses formulated in this study are as follows:

**Hypothesis** **1 (H1).**
*If the analyzed project genuinely promotes inclusion, it will achieve high scores in all four dimensions of inclusion: community, values, policies, and practices.*


**Hypothesis** **2 (H2).**
*No significant differences in inclusion scores will be found based on sex, gender, age, tenure, or role played in the project.*


**Hypothesis** **3 (H3).**
*Significant differences in inclusion scores will be found based on groups with and without disabilities.*


**Hypothesis** **4 (H4).**
*Significant differences will also be found based on special educational needs and physical or mental health problems.*


These hypotheses will serve as a theoretical framework for the analysis of the collected data, and will contribute to expanding our understanding of the perception of inclusion in non-formal or adult education contexts.

## 2. Materials and Methods

An ex post facto study of an exploratory and correlational nature has been conducted to analyse the relationships between the variables of interest derived from the population that has received the questionnaire used. Attending to [36], this design entails the analysis of pre-existing data, rather than the manipulation of variables, rendering direct causal conclusions impractical. While associations and patterns can be identified within the data, careful consideration is required when interpreting them as causal relationships. Notwithstanding its limitations, this design proves advantageous for exploring relationships in contexts where controlled experiments are infeasible. This approach provides a basis for future research, and may help to generate new hypotheses, and help us to better understand the phenomena studied around inclusive activities in a context with socially heterogeneous participants, with and without functional diversity.

### 2.1. Context and Sample of Participants

The study sample consisted of 135 active participants who are part of an inclusive choir based at the university of Zaragoza, with a track record of 6 years. This choir adopts an approach in which choral practice is seen as a means of personal development, transformation, and social inclusion. The composition of the choir is diverse, including people with and without functional diversity, and people of different ethnicities and nationalities. In addition, the choir includes the entire university educational community, as well as citizens with no previous or current relationship with the university.

All participants were invited to take part in the study during their regular choir rehearsal sessions. The invitation was extended to all active choir members, allowing them the opportunity to participate voluntarily. This approach ensured that a diverse and representative sample of choir members had the chance to contribute their insights and perspectives to the study (see Table 1).

Of the total of 153 choralists, 135 voluntary responses were obtained, and these can be considered as representative of the community as a whole (representing 88.2% of the total). The different profiles of participants include organisers (3), collaborators (5), and direct users (127). Of the sample, 65.2%, i.e., 88 people, are women; compared to 45 men; and 2 genderfluid people.

Of the participants, 62.2%, i.e., 84 people, are aged between 18 and 25; 15.6% are between 26 and 36 years old; 5.2% are between 37 and 50 years old; and 17% are over 51 years old. In terms of permanence, 41 people, i.e., 30.4%, had been involved in the project for less than 6 months, 8.9% between 6 months and 1 year, 20% between 1 and 2 years, 8.1% between 2 and 3 years, 8.9% between 3 and 4 years, 10.4% between 4 and 5 years, and 13.3% between 5 and 6 years.

Of the participants, 85.2% were from Spain, but there were also participants from Argentina, Brazil, Bulgaria, Canada, Chile, China, Ecuador, France, Ghana, India, Italy, Peru, Romania, Republic of Korea, and the United States. Regarding religion, 25.9% of the participants declared themselves agnostic, 27.4% atheist, and 32.6% Catholic Christians, and the remaining 14.1% declared themselves as belonging to other branches of Christianity, preferred not to say, or other. In terms of academic level, 7 participants have a doctorate, 7 have a master’s degree, 5 have a postgraduate degree, 33 have a university degree, 9 have a bachelor’s degree, 6 have a diploma, 17 have a professional qualification, 25 are currently studying at the university, 19 have a baccalaureate, 5 finished secondary education, and 2 finished primary education.

Of the participants, 22.2% have a recognised disability, including 9 people with a visual disability, 4 with a hearing disability, 6 with a motor disability, 4 with an intellectual disability, and 1 with a visceral disability. Of the participants, 9% were people with specific educational support needs, including 3 people with autism spectrum disorder or Asperger’s syndrome, and 8 people with ADHD. Of the 23% of participants who reported physical or mental health problems, 9 were people with anxiety disorder, 5 with depression, 3 with obsessive compulsive disorder, 3 with stage fright, and 1 with panic disorder.

### 2.2. Instrument

The instrument used to collect data in this study was a questionnaire adapted from the Evaluation of Inclusion Scale (SAI) [15]. This scale consists of 24 items grouped into four factors, developed from the Index for Inclusion, specifically designed for use in centres for people with disabilities that offer non-formal education. Each item in the scale is assessed on a 4-point Likert-type scale, where higher scores indicate more inclusive outcomes. This instrument is one of the few available to assess inclusion in this type of organisation. It has been selected in order to measure the level of social inclusion in the choir, even though it is not an organisation as such. It is easy to fill out, is designed for the same purpose as this work, and there is evidence of its validity and reliability.

In the study by [15], they provide evidence of the reliability and validity of the instrument, with internal consistency values above 0.96, and a validity analysis of its internal structure, through an exploratory factor analysis. In this analysis, the authors found four factors (see Table 2): community, composed of 6 items that assess the feeling of belonging to a group to which all members contribute, and where they are important; values, composed of 5 items that assess the existence of values focused on the person and their potential, as each person is unique; policies, made up of 6 items that assess the existence of policies and strategies that promote the defence of the rights of all people, diversity, and equality, as well as the promotion of people as unique individuals; and practices, made up of 7 items that assess actions based on the principle that each person has the right to have a personal project, to achieve their aspirations and goals, and to make decisions about their future.

The questionnaire used in this study was an adapted version, in which three of the items were removed from the original version (see Table 2), due to their irrelevance in the context of this study. Specifically, items 3, 17, and 21 were removed, as they were not relevant to the study. All items were to be answered via 4-level Likert-type response options (Strongly Disagree, Disagree, Agree, and Strongly Agree), identical to the original version. In addition to the adapted version of the SAI, 10 questions related to the socio-demographic assessment of the participants were included. The comprehensive questionnaire can be found in Appendix A.

The adaptation of the questionnaire, as implemented in this study, was undertaken to ensure its alignment with the specific research context and objectives. Within this framework, the decision to modify the questionnaire was guided by a careful consideration of its relevance to the study’s scope. It is essential to highlight that the Likert-type response format was maintained to ensure consistency with the original version, and facilitate comparative analysis.

### 2.3. Procedure

The procedure included three sessions with the research participants. In the first session, the study and its objectives were presented and explained, and the participants were asked for their informed consent [37]. In the second session, the instrument was applied, starting with the professionals in charge, followed by the collaborators and the participants. Finally, in the third session, the results were returned, in compliance with the ethical duty of social research to inform and involve the informants implicated in the process [38].

The sessions were conducted as a voluntary collective activity, with the entire group participating together in a shared classroom setting, adhering to the choir’s customary rehearsal schedule. Each participant was provided with a personal and individual device, allowing for privacy and tailored adaptation, where necessary. The timing of the questionnaire administration was deliberately positioned prior to engaging in the singing activities, affording participants adequate opportunity for contemplation and deliberate response. To facilitate the proceedings, the presence of three researchers ensured guidance and assistance throughout the sessions.

Each session lasted approximately one hour. The instrument was presented, and instructions for its completion were provided. During the sessions, it was necessary to read each indicator, and provide additional explanations based on the statements in the questionnaire. Support staff was available to provide physical assistance if necessary, as well as additional help to ensure understanding of the items and possible responses, taking into account the diversity of the group.

To ensure data confidentiality, several measures were implemented. The responses collected through Google Forms were stored on a password-protected and secure online platform. Access to the collected data was restricted to the research team members only. Additionally, all data were anonymised during the analysis phase, using unique identifiers instead of participants’ personal information.

During Session 1 of the study, participants were provided with a detailed explanation of the research objectives, procedures, and potential risks. They were then given the opportunity to ask questions and seek clarifications. Participants who agreed to take part in the study were asked to provide their informed consent by signing a consent form. This form outlined their rights as participants, and indicated their voluntary participation. The signed consent forms were securely stored, and kept separate from the collected data, to further ensure confidentiality.

### 2.4. Data Analysis

Although the SAI scale is initially supported by content validity [15], which, in turn, is supported by the literature review on inclusion and its components, as well as expert consultation and content analysis of each of its items, it is relevant to analyse the validity and reliability of the scale adapted for this study, given the modifications made to adapt it to the research context. With regard to the reliability of the scale used, the data were analysed using Jamovi software version 2.4.1. [39]. The acceptability of reliability was determined using the guideline provided by [40], where values exceeding 0.65 are considered acceptable under research circumstances. To assess the internal consistency of the scale, Cronbach’s alpha coefficient and McDonald’s omega coefficient were calculated. An assessment of construct validity was also carried out by means of an exploratory factor analysis, specifically using the principal component analysis method with Varimax rotation and Kaiser standardisation (following the same procedure used by the original authors of the instrument). This data analysis provides the basis from which to consider the factors or subscales of the SAI Scale as variables in subsequent analyses.

Given the size of the sample used in this study, it was decided to recode the variables of interest. It was observed that the detailed information collected generated categories with a reduced representation of participants. Considering this situation, it was decided to recode the variables into dichotomous categories. This facilitated the analysis and interpretation of the results, improving the representativeness of the groups in the study.

In order to respond to the main objective of this article, and to analyse the perceived level of inclusion in the music group and the factors associated with it, statistical analyses were carried out using both descriptive and inferential bivariate statistics (ANOVA) and Student’s *t*-tests, to test the hypotheses proposed, as well as the Jamovi version 2.4.1. programme [39]. To test the proposed hypotheses concerning the role of various variables (i.e., sex, gender, age, tenure, role played, etc.) in the level of inclusion and its factors, ANOVAs were employed for comparisons involving variables with more than two groups. Additionally, after the recoding of these variables into dichotomous categories, *t*-tests were applied. A significance level of 0.05 was adopted for these analyses.

### 2.5. Ethical Statement

In accordance with the ethical policies of the journal, the utmost responsibility and respect were maintained for the participants during the research, ensuring adherence to the strictest ethical standards. Prior to their participation in the study, informed consent was obtained from all subjects. Throughout the investigation, the confidentiality and anonymity of the participants were guaranteed, in alignment with the ethical principles of respect, beneficence, non-maleficence, return of results, and justice. Additionally, all the protocols suggested by the academic literature and the Declaration of Helsinki in relation to research with human subjects were rigorously followed [37].

Furthermore, it is noteworthy that the Cantatutti Inclusive Choir, where the research was carried out, operates under the supervision of the Music and Inclusion for Social Change Chair, which was established through an agreement between the University of Zaragoza and the Institute of Social Services of the Government of Aragon. This collaboration emphasises a commitment to integrity and ethics in research, especially in initiatives that promote social inclusion through music, and focus on the wellbeing and inclusion of vulnerable and disadvantaged individuals.

Given that the University of Zaragoza does not have a specific ethics committee, special care was taken to ensure compliance with the ethical guidelines set forth by the Music and Inclusion for Social Change Chair, which includes ethical research as one of its key objectives and purposes.

## 3. Results

### 3.1. Evidence of Reliability and Validity of the Adapted SAI Scale

Firstly, the reliability of the scale and the quality of its items were analysed using Cronbach’s alpha and McDonald’s omega. All items obtained values above 0.300 in the item–total correlation [41], and the scale as a whole has optimal reliability values (Table 3).

Secondly, an exploratory factor analysis was conducted, using the principal components method, with Varimax rotation and Kaiser standardisation, to assess construct validity, following a procedure identical to that used by the original authors of the SAI [15]. The results were arranged according to four factors (Table 4), as in the SAI, in order to achieve the greatest similarity with that scale, being able to explain 53.6% of the variance.

Table 5 provides the rotated component matrix in which the factor loadings of the items on the different factors can be observed. Although the factorial solution is not clear-cut, it is observed that items have significant factor loadings (above 0.300) on more than one factor, and the factors obtained from the data, as well as the changes with respect to the original structure of the scale, are described below. Cronbach’s alpha value is also provided, to estimate the internal consistency of the factors as independent subscales.

Factor 1, called “Values”, consists of five items that focus on personal values, and the recognition of each person’s individual abilities. It emphasises belief in the unique potential of each individual, and the importance of having confidence in their abilities. Four of the five items that were originally part of this factor are retained. One of them (item VII) previously belonged to Values, and now becomes part of the Policies factor, and item X, originally belonging to Policies in the SAI, is now part of the Values factor. The subscale obtained a Cronbach’s alpha coefficient of 0.752.

Factor 2, called “Practices”, consists of five items that refer to actions based on the principle that each person has the right to have personal goals and aspirations, as well as to make decisions about their future. Item XVII is retained in this factor (despite having a similar factor loading also in another factor) for consistency with the initial SAI scale. The subscale obtained a Cronbach’s alpha coefficient of 0.768.

Factor 3, called “Policies”, is composed of six items that refer to the existence of policies and strategies that promote the defence of rights, diversity, equality, and the promotion of people as unique individuals. Items II and III are maintained in this factor (although they show similar factor loadings in another factor), for consistency with the SAI. However, items VII and XII, previously belonging to Values and Community, are incorporated into “Policies”, as their factor loadings are much higher in this one, with respect to those obtained in other factors. The subscale obtained a Cronbach’s alpha coefficient of 0.776.

Factor 4, called “Community”, consists of five items that refer to aspects related to belonging to a team in which all members are valued and play a relevant role in the proper development of practices. The importance of creating inclusive processes where everyone feels part of the team is highlighted. Item XIX is retained in this factor (despite also having a similar factor loading in another one) for consistency with the SAI. The subscale obtained a Cronbach’s alpha coefficient of 0.732.

In conclusion, the instrument, with its final configuration of 21 items (Table 6), has demonstrated evidence of reliability and validity, both overall, and on the different subscales.

Based on the results of the adapted SAI questionnaire, the mean scores of the participants, according to the factors, are shown in Table 7. It should be noted that the responses on this scale range from 1 to 4, with “Strongly Agree” being assigned a 1, and “Strongly Disagree” being assigned a 4. In this study, direct scores were calculated for the different factors assessed via the questionnaire used. Once these scores were obtained, typification was carried out via converting them into z-scores. This typing process made it possible to standardise the scores in relation to the mean and standard deviation of the sample, thus facilitating the comparison and contrasting of the results between the different factors and groups of interest. As can be seen in Table 7, the typed means indicate a higher perception of inclusion in the factor corresponding to “Values”, followed by “Community”, “Policies”, and “Practices”. The values are, in any case, very close to each other.

### 3.2. Relationships between the Variables of Interest

In the present study, data analysis was carried out in order to examine the relationships between the responses to the scale used in this research (and the factors assessed via the scale) and the variables of interest: “profile”, “length of participation”, “age”, “sex”, “gender”, “religion”, “country”, “academic degree”, “disability”, “special needs”, and “physical and mental health problems”. (The category “disability” encompasses conditions that result in impairments affecting various aspects of daily life functioning. The term “special needs” denotes requirements essential for individuals facing specific challenges, which may not necessarily align with a formally recognised disability category. “Physical and mental health problems” encompass conditions impacting a person’s physical wellbeing or mental health, which may not always directly correspond to a formally diagnosed disability. These distinctions were established based on participants’ self-reported information, and any available medical documentation).

In a preliminary analysis, the results revealed that most of the variables of interest (i.e., “profile”, “length of association”, “age”, “sex”, “gender”, “religion”, “country”, “academic degree”, “special needs”, and “physical and mental health problems”) did not show significant differences with the responses to the inclusion questionnaire. In contrast, statistical significance was observed for the ‘disability’ variable, indicating that there are statistically significant differences in perceptions of the level of inclusion in some of the factors of the scale between participants with and without disabilities. The preliminary analysis results (ANOVA) can be found in Appendix B.

No significant differences were observed in the analysed dimensions, or else comparisons within variables could not be conducted due to a limited number of participants in some of their categories. Given this circumstance, and the desire to conduct a more in-depth analysis, owing to the unequal sub-sample sizes, it was decided to recode all variables into dichotomous forms. These recoded variables were subsequently subjected to Student’s *t*-tests, leading to new significant findings among the respondent groups (Table 8). Consequently, through this approach, the data analysis revealed noteworthy distinctions in the variables “age”, “religion”, “academic degree”, “disability”, and “health problems”.

In order to explore the relationship between the variable “age” and the results for the different factors of the SAI scale, the pre-defined age categories 18–25 years, 26–36 years, 37–50 years, and over 51 years were initially examined. However, no significant differences were found between these categories in relation to the scale scores. The age variable was recoded into two broad categories, considering that more than 60% of the participants belonged to the group of 18–25 year olds, while the rest were older than 25. In this analysis, a significant relationship was found around factor 2, called “Practices” (Table 8). It was observed that those aged 25 and over had a more inclusive perception of the choir’s “practices” factor, compared to those in the 18–25 age range.

In the case of “religion”, the pre-established categories within the variable were preliminarily assessed. No significant relationships were found between these categories and the scale results. However, when a comparison analysis was carried out between participants who identified themselves as believers and those who identified themselves as non-believers, a significant relationship of the variable was identified around factor 2, “Practices”, of the inclusion assessment scale. It was found that people with religious beliefs had a higher perception of inclusion in choir practices than those who self-reported as non-believers.

As for “academic title”, initially, the pre-established categories within the variable were assessed to determine whether there were significant relationships with the scale scores. However, no substantial associations were found between these categories and the inclusion scores. A comparison analysis was then carried out between two groups: those participants with a university degree, and those without. In this analysis, a significant relationship was identified around factors 3 and 4 of the scale, i.e., “Policy” and “Community”, respectively (Table 8). It was observed that individuals without a university degree expressed a higher perception of inclusion in the choir’s policies and community compared to those individuals who had one.

With regard to the “disability” variable, particularly significant relationships were found between participants with and without disabilities and factors 2 and 3 of the inclusion scale, corresponding to “Practices” and “Policies”, respectively (Table 8). A notable association was found between the presence of disability and a more inclusive perception of the choir, in the aspects related to policies and practices.

The dimension of “physical and mental health problems” was assessed according to the pre-established categories within the variable. No significant relationships were found between these categories and the results of the scale. However, when a comparison analysis was carried out between participants with and without pathologies, a substantial relationship of the variable was identified around factors 3 and 4 of the scale, i.e., “Policies” and “Community” (Table 8). It was found that people without problems in these areas perceived a greater inclusion in the choir’s policies and community.

## 4. Discussion

The instrument used in this study has shown evidence of reliability and validity, partially coinciding with the original results of [15]. In terms of reliability, satisfactory Cronbach’s alpha coefficients were found for the total scores and individual factors, indicating an adequate internal consistency in the participants’ responses. In terms of validity, although partial modifications were made to the scale structure compared to the original version, the original four factors were retained, suggesting that the instrument still captures the essential aspects of inclusion assessment. This evidence of reliability and validity supports the use of the instrument in the present study, and provides a basis for the interpretation of the results. However, it would be interesting to further explore and validate the instrument in different contexts and populations, to gain a more complete understanding of its usefulness and applicability.

The results support the hypothesis posited in the study, which stated that “if the analysed project truly promotes inclusion, it was expected to receive high scores in all four dimensions of inclusion”.

Regarding the second hypothesis of this study, which posited that no significant differences would be found in inclusion scores based on sex, gender, age, tenure, or role played in the project, initially, the results supported this assertion for all variables of interest. However, upon grouping responses by age variable, a significant difference was observed between respondents under 25 years old and those over 26 years old.

The third hypothesis of this study asserted that significant differences in inclusion scores would be found in groups with and without disabilities. The obtained results support this assertion, as individuals with recognised disabilities rated inclusion with significantly higher scores compared to participants without disabilities.

Finally, with regard to the fourth hypothesis of this study, which posits that significant differences would be found based on special educational needs and physical or mental health problems, the results revealed a partial fulfilment of this hypothesis. No significant differences were found between individuals with and without special educational needs, indicating that this variable did not significantly influence the perception of inclusion. However, relevant differences were observed between individuals with and without physical or mental health problems, suggesting that this condition may impact perceptions of inclusion.

The main objective of this study was to analyse the perception of inclusion of the members participating in the Cantatutti Inclusive Choir, taking into account some of the variables of interest that characterise their diversity, including whether or not they have a disability. For this purpose, an inclusion assessment scale has been used, covering four factors: Practices, Community, Policies, and Values [15].

Significantly, the mean scores obtained for all factors are very low, indicating that inclusion is assessed mostly positively. These results support the approach of [35], which stated that if the project under analysis truly promotes inclusion, it would be expected to obtain overtly positive scores on all four dimensions of inclusion. The strategies and policies implemented in this choir favour the rights, diversity, equality, and promotion of its members. These findings are consistent with previous research [30,31,42], which highlights the importance of projects where music plays a key role as a working tool.

It is also relevant to note that there is a minimal difference between the average scores of the different factors, with the perception of the project’s Values standing out positively. Of the four factors analysed, the Values factor obtained a higher score, possibly due to the fact that it is a fundamental element around which Policies, Practices, and the Community are articulated. These results do not coincide with those obtained in [35], for whom the Practices factor scored higher, and represents the most visible and tangible element of inclusion, considering that practices concretise policies and cultures. The Values factor obtained a relatively low score, which the authors attribute to its more abstract and implicit nature, making it difficult to assess accurately. In this respect, it would be interesting to produce new research based on the SAI that would yield new results for comparison and discussion.

Within the framework of the study, it has been observed that values play a key role, and score highly. These values should be focused on people and their individual abilities, especially valuing a climate of respect and acceptance towards the diversity of the members, as documented in previous research [29]. From the moment they join the choir, it must be ensured that no one feels excluded, regardless of any personal, social, or cultural differences that may exist. This issue is of great relevance, as it highlights the need for values that focus on individuals and their potential to promote inclusion within the choir [17].

On the one hand, a preliminary analysis of the data shows no significant differences in the inclusion scores based on sex, gender, age, length of association, or role in the project. However, when recoding the age variable, a significant difference was observed between respondents under 25 and those over 25. The fact that those aged 26 and over identified the ‘practices’ as more inclusive compared to those aged 18–25 may be attributed to a possible generational influence. It is plausible to consider that younger individuals perceive inclusion more naturally, due to their exposure to a social environment that has evolved towards more inclusive practices compared to those who, through their experience, have witnessed the transformation of society from times when inclusive practices were less prevalent, or exclusively associated with the concept of integration [2].

On the other hand, the results show that people with a recognised disability perceive inclusion more significantly compared to non-disabled participants. The tendency of people with a recognised disability to evaluate practices and policies more positively than non-disabled participants could be related to their previous experiences in activities that have been socially less inclusive [43]. These individuals may have experienced situations in which they have not had the same opportunities as their peers, both in terms of specific practices, and in relation to regulations and policies that seek to promote equity and inclusion in the educational environment [44]. It is, therefore, possible that their more favourable assessment reflects a contrast between past experiences of exclusion and the current inclusion measures implemented, in the context of the study.

Furthermore, no significant differences were found between people with and without special educational needs, indicating that this variable did not significantly influence perceptions of inclusion. However, notable differences were observed between people with and without physical or mental health problems, suggesting that this condition may have an impact on perceptions of inclusion. The observed tendency for people with physical or mental health problems, particularly those experiencing depression or anxiety, to give lower scores on the evaluation of inclusion in terms of ‘community’ and ‘policies’ is interpreted as related to their social vulnerability [45]. These results provide relevant information for the project in question, as they indicate the need to take specific measures to support these people, and promote their greater sense of inclusion within the group. By considering their vulnerable situation, the project can implement strategies and actions that address the specific barriers faced by these people, with the aim of fostering a greater sense of belonging and participation in the community.

According to [28], participation in these musical experiences can strengthen the sense of community and of belonging to a group. In the context of these activities, all members should have the opportunity to contribute to the ensemble, and feel valued as an important part of it. It is especially relevant to note that more than half of the participants in this study identify with various forms of disabilities, disorders, addictions, or illnesses. Through activities focused on strengthening choral activity and promoting socio-emotional development, choristers can find the support they need to make their individual contributions, which helps to foster their sense of inclusion and participation in the choral community.

The results reveal other significant findings, such as the fact that people without a university education score higher on the “policy” and “community” factors, compared to those with this kind of education. This phenomenon may be related, as in the case of disability, to the participants’ previous experiences, or to the fact that they feel included in a project hosted by a university institution, despite the fact that they did not have a university education. Equally, this result could also be related to the fact that people with a university education have developed a reflective and critical attitude towards the opportunities for improvement of an inclusive project. Their academic background and experience in the university environment might have provided them with tools to more thoroughly analyse and evaluate the inclusion policies and practices implemented in the project [46].

These results underline the importance of considering the diverse educational background and life experience of participants when assessing inclusion in a project. They also suggest the need to encourage participation and feedback from people with different educational backgrounds, as each group can bring valuable perspectives, to enrich inclusive policies and practices.

This study has some limitations that should be taken into account when interpreting the results. Firstly, the sample used may limit the generalisability of the findings, as it was based on a specific population, and does not necessarily represent all inclusive choir contexts. In addition, the research focused on participants’ self-perception, which could be subject to individual bias and variation. The study did not control some external variables that could influence perceptions of inclusion, such as the social and cultural environment. Therefore, it seems pertinent to conduct future research with larger and more diversified samples, as well as to incorporate new methods and data triangulation, in order to gain a more complete understanding of the phenomenon of inclusion in inclusive choirs.

Although the validation procedure of the original instrument has been rigorously replicated, the explained variance, standing at 53.5%, may be deemed relatively moderate in specific contexts. Consequently, judicious consideration is recommended. To bolster the instrument’s reliability and analytical depth, future research could replicate these findings across diverse cohorts, and explore advanced statistical methodologies, such as structural equation modelling. These limitations underscore the imperative for ongoing validation and inquiry into the instrument’s efficacy, to ensure its resilience and relevance across varied scenarios.

It is important to acknowledge the limitations associated with converting responses from instruments utilising categorical response formats into numerical values. In this study, the commonly observed approach employed in analogous research was adhered to, to address this format. However, caution is advised when directly transforming Likert ratings into numerical values. It is crucial to recognise that assuming an equidistant step size between labels, with a difference of 1, may not hold true unless explicitly stipulated within the questionnaire itself. The intricacies inherent in response scaling underscore the significance of employing a well-considered methodology that respects the nuanced nature of response categories.

Finally, for [35], it is of the utmost importance to assess the fundamental dimensions of inclusion, taking into consideration the perception of all members involved in the community. Future research, using complementary methodologies, such as in-depth interviews or focus groups, can play a key role in providing further insights into this complex and evolving phenomenon.

## Figures and Tables

**Table 1 behavsci-13-00758-t001:** Technical data sheet of the sample.

Technical Specifications	
Scope	City of Zaragoza
Group	Cantatutti Inclusive Choir
Total choir members	153 choristers
Sample assessed	135 choristers
Method	Google Forms sent via WhatsApp
Date	April 2023

**Table 2 behavsci-13-00758-t002:** Changes made to the SAI for the final version of the scale (number of questions).

Factors	SAI [15]	Adapted Final
Practices	7	5
Community	6	6
Policies	6	5
Values	5	5

**Table 3 behavsci-13-00758-t003:** Scale reliability statistics.

Cronbach’s α	McDonald’s ω
0.899	0.902

**Table 4 behavsci-13-00758-t004:** Summary of principal components.

Component	SC Loadings	% of Variance	Accumulated % Cumulative
1	3.21	15.3	15.3
2	2.69	12.8	28.1
3	2.69	12.8	40.9
4	2.67	12.7	53.6

**Table 5 behavsci-13-00758-t005:** Component loading.

		Component			
Item-Factor (Originals)	1	2	3	4	Uniqueness
I—Policies			0.791 *		0.348
II—Policies			0.333 *	0.312	0.652
III—Policies			0.505 *	0.514	0.453
IV—Policies	0.543		0.574 *		0.363
V—Values	0.662 *				0.444
VI—Values	0.585 *				0.619
VII—Values			0.534 *	0.347	0.553
VIII—Values	0.533 *				0.584
IX—Values	0.727 *				0.417
X—Policies	0.693 *				0.429
XI—Practices	0.332	0.670 *			0.390
XII—Community	0.300	0.309	0.627 *		0.421
XIII—Community	0.302			0.629 *	0.458
XIV—Community	0.374			0.535 *	0.518
XV—Practices		0.795 *			0.326
XVI—Practices		0.766 *			0.351
XVII—Practices		0.445 *		0.456	0.518
XVIII—Practices	0.404	0.410 *			0.557
XIX—Community		0.437	0.412	0.416 *	0.462
XX—Community		0.322		0.630 *	0.423
XXI—Community			0.344	0.603 *	0.456

Note: Varimax rotation was used. Note: * items that are part of the new adapted factor.

**Table 6 behavsci-13-00758-t006:** Relationship of questions associated with factors in the original and adapted SAI scale.

Factor-Questions (Original)	Factor-Questions (Adapted)
I—Policies	I—Policies
II—Policies	II—Policies
III—Policies	III—Policies
IV—Policies	IV—Policies
V—Values	V—Values
VI—Values	VI—Values
VII—Values	VII—Policies *
VIII—Values	VIII—Values
IX—Values	IX—Values
X—Policies	X—Values *
XI—Practices	XI—Practices
XII—Community	XII—Policies *
XIII—Community	XIII—Community
XIV—Community	XIV—Community
XV—Practices	XV—Practices
XVI—Practices	XVI—Practices
XVII—Practices	XVII—Practices
XVIII—Practices	XVIII—Practices
XIX—Community	XIX—Community
XX—Community	XX—Community
XXI—Community	XXI—Community

Note: * modified factors.

**Table 7 behavsci-13-00758-t007:** Factors and descriptives.

	Direct Scores				Typified Scores		
Factors	Mean	SD	Minimum	Maximum	Mean	Minimum	Maximum
Values	6.44	1.64	5	11	1.298	−0.880	2.77
Practices	7.45	2.08	5	13	1.490	−1.181	2.67
Policies	8.61	2.28	6	17	1.430	−1.142	3.68
Community	7.15	1.93	5	13	1.428	−1.111	3.03

**Table 8 behavsci-13-00758-t008:** Analysis of dimensions (*t*-tests).

Dimension	Factors	Groups	Mean	SD	*t*	*p*
Age	Values	18–25 years	0.08458	0.979	1.264	0.208
		>25 years	−0.13931	1.029		
	Practices	18–25 years	0.13789	1.028	2.0816	0.039 *
		>25 years	−0.22711	0.916		
	Policies	18–25 years	0.11984	1.034	1.8019	0.074
		>25 years	−0.19738	0.916		
	Community	18–25 years	−0.00274	0.99	−0.0406	0.968
		>25 years	0.00451	1.026		
Religion	Values	Non-believers	−0.04229	0.983	−0.524	0.601
		Believers	0.0483	1.025		
	Practices	Non-believers	0.20384	1.045	2.585	0.011 *
		Believers	−0.233	0.899		
	Policies	Non-believers	0.03813	0.949	0.472	0.638
		Believers	−0.0436	1.061		
	Community	Non-believers	0.00958	1.016	0.119	0.906
		Believers	−0.0109	0.99		
Academic degree	Values	University students	0.0603	1.019	1.03	0.307
		Not university st.	−0.129	0.957		
	Practices	University students	0.0756	0.992	1.29	0.2
		Not university st.	−0.162	1.009		
	Policies	University students	0.1291	0.992	2.23	0.028 *
		Not university st.	−0.276	0.973		
	Community	University students	0.1257	1.037	2.17	0.032 *
		Not university st.	−0.269	0.867		
Disability	Values	With disab.	−0.0677	1.041	−0.419	0.676
		No disab.	0.0193	0.992		
	Practices	With disab.	−0.5228	0.805	−3.37	<0.001 ***
		No disab.	0.1494	1.003		
	Policies	With disab.	−0.4413	0.802	−2.811	0.006 **
		No disab.	0.1261	1.018		
	Community	With disab.	−0.2662	0.946	−1.664	0.098
		No disab.	0.0761	1.006		
Health problems	Values	No problem	−0.0013	0.974	−0.0276	0.978
		With health problem	0.00437	1.1		
	Practices	No problem	−0.07408	0.915	−1.5854	0.115
		With health problem	0.24852	1.23		
	Policies	No problem	−0.11024	0.87	−2.3869	0.018 *
		With health problem	0.36984	1.3		
	Community	No problem	−0.1164	0.947	−2.5263	0.013 *
		With health problem	0.39049	1.09		

Note: * *p* < 0.05, ** *p* < 0.01, *** *p* < 0.001.

## Data Availability

Data are available upon request to the corresponding author.

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
