# Peer review of "Self-Perception of Inclusion in an Inclusive Choir: An Analysis Using the Scale for the Assessment of Inclusion (SAI)"

_behavsci, 2023, doi:10.3390/bs13090758_

Round 1

Reviewer 1 Report

This paper is well-written and I enjoyed reading it. I believe there is a need for more papers exploring these topics.

Overall, the manuscript is rather easy to follow, but I list some suggestions for how the paper could be made stronger and how readability could be improved, below. 

From what I understand, the data is not shared. This makes it a little bit difficult to reproduce/verify the results. It would be good if the numbers could be shared. This also goes for the questionnaire. I believe that it is important to have access to the entire questionnaire to fully understand the dataset.

It would be good if the results discussed in the text would immediately point the reader to a specific table so that it is easy to see where the nonsignificant versus significant results are found (upon first mentioning of such results, at the top of the section). See e.g. line 412. 

It would also be easier for the reader to follow the analysis if it was more clear when the respective statistical method was used (see section 3.2 - "preliminary analysis"). Currently, it is not entirely clear when e.g. ANOVA was used. Is the test for all (original categories) done with the ANOVA and t-tests used for the groupings when categories were merged? From what I understand the ANOVAs are in Table 8.

The reference section is rather substantial and covers important state of the art. However, relatively little is said about previous/related work on inclusive choirs and inclusive music ensembles in general: is there perhaps more to be said about this? How has the concept of "inclusion" been evaluated in other ensembles in other countries? 

The "comprehensive instrument" is not really defined. How is this different from the "questionnaire"? What does it mean that "the instrument was applied"? The "instrument" is mentioned on line 61 but the meaning is not entirely clear.

Why was 4 point Likert Scale used (not more steps)?

Were the sessions conducted in groups? Or individually? Directly after singing? How many researchers were there to help out with the assessment?

53.5% variance explained, is that not a somewhat small number?

Some advice against transforming Likert ratings directly into numerical values since it is not necessarily true that the step size between labels is 1 (unless this is explicitly stated in the questionnaire).

Line 389: the following terms are not explained/defined in detail: profile & length of association. In addition, it is not entirely clear how the distinction between "disability", "special needs", and "physical and mental health problems" is made.

Finally, it would be very nice to see and hear music examples of the choir, if possible.

Minor comments: 

This is probably the LaTeX table but some tables are broken on two pages: it would be better if they would be on the same page. 

Some table captions would need some more information. 

Line 17 in the abstract: when I read this the first time I understood it as though being part of an inclusive choir, in general -> leads to x,y,z (meaning that there were data points from people in different choirs). It would perhaps be good to highlight that what is meant here refers only to one specific choir in Spain and that no comparisons between choirs are made. 

I marked "ethical concerns" mainly because I was confused about the reasoning that the university in question did not have an ethics board -> no ethics approval. Legislations are different in different countries, ofc, but in many European countries, one is required to get ethics approval from a National Board if there is none at the university level. If specific rules exist in Spain which make you exempt from obtaining ethics approval (even when working with subjects with disabilities), this should be clearly explained. 

Line 67: here "Spanish" is mentioned a little bit out of the blue, but it is not clear before that the study is focusing specifically on a choir in Spain. 

Line 202: which "community" are you referring to here?

Table 2: Are the numbers referring to the "number of questions"?

Line 264: What is the difference between "collaborators" and "professionals"

This paper is well written overall and there are only minor details to be fixed in terms of language. 

Table 1: Universe -> University. 

Lines 80-83: long, needs revision. 

Line 105: it would be good to add the country in which this university is located

Line 108: since -> active since?

Line 148: full stop in wrong place?

Line 178: tries -> tries to 

Line 205: Fluid gender - is this how these people defined themselves?

Line 305: justice.. -> justice. 

Reviewer 2 Report

I see this paper as a relevant and interesting in a field of inclusion (and inclusived education) thanks to the (specific)  topic, whis has not been deeply described in many studies until now, and which is focused outside the narrow field of (school) education. Perhaps it was not necessary to present all of well known information regarding the history and evolution of the inclusive education... Of course, there are the serious limits of this research study (mentioned in the paper), and I also consier some interpretations of results in a Discussion section as too simplifying - however, it was relevant in the content and it is not a reason to rewrite the whole text at all. I find this paper a good contribution to discussion on the inclusive concept today.

Reviewer 3 Report

TITLE AND ABSTRACT

In general, the title and abstract of the scientific article are informative and well-structured, presenting the focus of the study and its objectives clearly. The abstract provides an overview of the research design and key findings. However, there is room for improvement.

Abstract: The abstract needs to be clearer and more concise. It should include specific information about the study objectives, the sample characteristics, key results, and main conclusions. Additionally, a brief description of the methods used could be added.

INTRODUCTION

The introduction of the article provides a comprehensive overview of the concept of inclusion, its evolution, and its application in various contexts, particularly in the field of education and music. However, there are some areas where the introduction could be improved:

-          Clarity and Conciseness: The introduction is quite lengthy and includes several repetitions and redundancies. It could be condensed to focus on the key points and central objectives of the study.

-          Specific Research Gap: While the introduction provides a background on inclusion and its significance, it lacks a clear statement of the specific research gap or problem that the study aims to address. A clearer articulation of the research question or objective would enhance the focus of the introduction.

-          Connection to Previous Literature: The introduction briefly mentions some studies on the benefits of music-social projects for social inclusion, but it could be strengthened by providing a more comprehensive review of relevant literature on inclusive practices in non-formal education and their impact.

-          Justification for the Adaptation: The introduction introduces the Scale for the Assessment of Inclusion (SAI) as an adapted version of the Index for Inclusion. However, it would be helpful to provide a stronger justification for the adaptation and explain how the SAI specifically addresses the needs of people with disabilities and the organizations serving them.

-          Hypotheses or Research Questions: The introduction could benefit from explicitly stating the research questions or hypotheses that the study seeks to investigate. This would provide readers with a clear understanding of the study's aims and focus.

Suggested Improvements:

1.       Condense the introduction by removing redundancies and unnecessary details.

2.       Clearly state the specific research gap or problem that the study aims to address.

3.       Provide a more comprehensive review of relevant literature on inclusive practices in non-formal education and music-social projects.

4.       Strengthen the justification for the adaptation of the SAI and highlight its relevance to the study's context.

5.       Explicitly state the research questions or hypotheses that the study seeks to investigate.

MATERIALS AND METHODS

The "Materials and Methods" section provides essential information about the study's design, sample, instrument, and data analysis. However, there are some areas where clarity and additional details would enhance the section:

-          Study Design: The section mentions that the study is of an ex post facto nature, but it does not provide a clear explanation of this design. Including a brief description of what an ex post facto study entails and its implications for causality would be beneficial.

-          Sample Description: While the section provides basic demographic information about the participants, it would be helpful to elaborate on the recruitment process and inclusion criteria for the choir members. Providing information on how participants were invited to take part and the sampling procedure used would increase the transparency of the study.

-          Data Collection: The section briefly mentions that the instrument was administered via Google Forms sent via WhatsApp, but it lacks details on how data confidentiality and informed consent were ensured. Adding information about the informed consent process and data privacy measures would be appropriate.

-          Data Analysis: The section mentions the use of Cronbach's alpha and McDonald's Omega coefficients for reliability assessment and principal component analysis for construct validity. However, it would be beneficial to specify the cutoff values used to determine acceptable reliability and how the factor analysis was performed (e.g., number of factors retained).

-          Recoding of Variables: The section mentions recoding variables into dichotomous categories but does not explain the rationale behind this decision. Providing a justification for the recoding process and any potential implications for the analysis would improve transparency.

-          Statistical Analysis: While the section mentions the use of ANOVA and t-tests, it would be helpful to specify the hypotheses tested and the significance level used for inferential analyses.

Suggested Improvements:

1.       Provide a brief explanation of the ex post facto study design and its implications for the study.

2.       Elaborate on the recruitment process and inclusion criteria for selecting choir members.

3.       Include information about the informed consent process and data privacy measures for data collection.

4.       Specify the cutoff values for acceptable reliability, explain the number of factors retained in the factor analysis, and clarify the rationale for recoding variables into dichotomous categories.

5.       Clearly state the hypotheses tested and the significance level used for inferential analyses.

RESULTS

The "Results" section presents the findings of the study and explores the relationships between the adapted SAI scale and the variables of interest. Overall, the results are well-presented and informative, but there are a few areas that could be improved:

-          Factor Analysis Interpretation: The factor analysis results are briefly summarized, but the interpretation of the factors and their relationship to the original SAI scale could be expanded. Discussing the similarities and differences in factor loadings between the adapted and original versions of the scale would provide more insights.

-          Clarity on Analysis Procedure: The recoding of variables is mentioned, but the rationale behind the recoding decisions is not explained. Providing a clear justification for the recoding procedure will help readers understand why certain categories were merged.

Suggested Improvements:

1.       Detailed Factor Analysis: Provide a more comprehensive interpretation of the factor analysis results. Discuss why certain items were grouped together under specific factors and how they relate to the conceptual framework of the adapted SAI scale.

2.       Justify Recoding Decisions: Clarify the rationale behind the recoding of variables into dichotomous categories. Explain why certain categories were combined and how this process improves the analysis.

DISCUSSION

The "Discussion" section of the article provides a comprehensive analysis of the study's findings. It effectively links the results to the research objectives and relevant literature. However, there are a few areas that could be further improved:

-          Interpretation of Results: While the discussion highlights significant findings related to different variables, it could benefit from more in-depth interpretation of the implications of these findings. Providing a deeper analysis of why certain relationships were observed and how they align with theoretical frameworks would strengthen the discussion.

-          Addressing Limitations: The discussion mentions some limitations of the study, but it could be more explicit in discussing how these limitations may impact the generalizability and validity of the findings. Discussing the potential biases and their implications will add nuance to the interpretation of the results.

Suggested Improvements:

1.       In-depth Interpretation: Go beyond describing significant findings and provide a more extensive analysis of why certain relationships were observed. Relate the findings to theoretical frameworks or practical implications to offer a deeper understanding of the results.

2.       Address Limitations Adequately: Elaborate on the study's limitations and discuss how they may have influenced the results. Be transparent about potential biases and their impact on the validity of the findings.

By implementing these improvements, the "Discussion" section will become more robust and insightful, providing a comprehensive analysis of the study's findings and their implications for the field.
